# Population Pharmacokinetics of Difloxacin in Crucian Carp (*Carassius auratus*) after a Single Oral Administration

**DOI:** 10.3390/vetsci10070416

**Published:** 2023-06-27

**Authors:** Kai-Li Ma, Fang Yang, Mei Zhang, Jun-Cheng Chen, Ming-Hui Duan, Ze-En Li, Yan Dai, Yue Liu, Yang-Guang Jin, Fan Yang

**Affiliations:** College of Animal Science and Technology, Henan University of Science and Technology, Luoyang 471023, China

**Keywords:** crucian carp, difloxacin, population pharmacokinetics, sparse sampling, oral dosing

## Abstract

**Simple Summary:**

Crucian carp (*Carassius auratus*) is a freshwater fish that is popular due to its high nutritional value and delicious taste. It has become one of the most commonly farmed fish species, particularly in China. With the rapid development of intensive aquaculture, fish are becoming increasingly vulnerable to various pathogens that can cause significant economic losses. To reduce these losses, treatment with effective antibacterials is necessary. Although not approved in China, difloxacin is being used extensively in an extra-label manner to treat infections in aquaculture. However, the dosage guidelines specific to fish have not been established, resulting in the extrapolation of mammalian dosing regimens. Because fish are poikilotherms with unique physiologies and anatomies, dosing regimens intended for mammals may not be appropriate for them. In this research paper, we have employed a sparse sampling method and non-linear mixed effect modeling to develop population pharmacokinetics models. In addition, we have determined the values of the PK/PD parameter (free AUC/MIC) by utilizing previously published MIC data. Based on the free AUC/MIC values calculated herein, the current oral dose of difloxacin (20 mg/kg body weight) might be not enough to treat infections in crucian carp.

**Abstract:**

This study aimed to investigate the population pharmacokinetics of difloxacin in crucian carp (*Carassius auratus*) orally provided a single dose of 20 mg/kg body weight (BW). To achieve this, fish were sampled at various intervals using a sparse sampling strategy, and plasma samples were analyzed using the high-performance liquid chromatography (HPLC) method. Subsequently, naïve average data were analyzed using a non-compartmental method, and a population model was developed based on the nonlinear mixed effects approach. The covariate of BW and the relationship between covariances were sequentially incorporated into the population model. However, it was found that only covariance and not BW affected the population parameters. Therefore, the covariance model was taken as the final population model, which revealed that the typical values of the absorption rate constant (tvKa), apparent volume of distribution per bioavailability (tvV), and clearance rate per bioavailability (tvCl) were 1.18 1/h, 14.18 L/kg, and 0.20 L/h/kg, respectively. Based on the calculated free AUC/MIC values, the current oral dose of difloxacin (20 mg/kg BW) cannot generate adequate plasma concentrations to inhibit pathogens with MIC values above 0.83 μg/mL. Further study should be carried out to collect the pathogens from crucian carp and determine the MIC data of difloxacin against them. Pharmacodynamic experiments must also be further carried out to determine the optimal therapeutic dose for the treatment of *Aeromonas hydrophila* infection.

## 1. Introduction

Crucian carp (*Carassius auratus*) is a popular freshwater fish due to its delicious taste and high nutritional value [1]. As an adaptable species, it can be cultured in various regions, which makes it accessible worldwide [2]. In the past few decades, fish have become a crucial source of animal protein and nutritious food for the global human population [3]. In response to the growing demand for fish, the aquaculture industry has expanded significantly [4]. However, this growth has also led to the emergence of bacterial diseases which can cause significant economic losses for farmers [5]. To mitigate these losses, treatments with effective antibacterial agents become necessary.

Fluoroquinolones are a highly important class of antibiotics, ranking third in worldwide usage due to their broad-spectrum antimicrobial activity, low resistance rates, and cost-effectiveness [6]. In aquaculture, these antibiotics are commonly utilized to treat various infections, including respiratory, urinary, and digestive tract infections, as well as septicemia [7,8]. Their bactericidal effects are concentration-dependent and result from their inhibition of bacterial DNA topoisomerases II and IV [6].

Difloxacin, a third-generation fluoroquinolone, possesses similar antimicrobial effects to other members of this class [9]. With a low MIC value against susceptible fish pathogens and efficient distribution upon oral administration, difloxacin has proven to be an effective treatment for systemic bacterial infections in fish [9]. Although not approved in China [10], difloxacin is being used extensively in an extra-label manner to treat infections in aquaculture [11]. However, dosage guidelines specific to fish have not been established, resulting in the extrapolation of mammalian dosing regimens. Because fish are poikilotherms with unique physiologies and anatomies, dosing regimens intended for mammals may not be appropriate for this species [12].

Pharmacokinetics is a powerful tool in determining dosing regimens. The pharmacokinetics of difloxacin have been studied in several fish species, including gibel carp [9], crucian carp [1], olive flounder [13], grass carp [14], and Atlantic salmon [15]. However, all previous studies were conducted via traditional methods involving large numbers of aquatic subjects but yielding limited profiles. Those traditional pharmacokinetics studies allowed for the calculation of pharmacokinetic parameters using average concentration–time data. As a result, inter-individual variations were not accounted for in these previous studies. To date, these approaches may be too simplistic to effectively model the complex variations in fish populations. Given these limitations, new strategies such as population pharmacokinetics modeling may be necessary to optimize dosing regimens and ensure the maximum efficacy of difloxacin in fish populations.

Population pharmacokinetics is an effective means of expalining the random effects of drugs on subjects [16,17]. It encompasses inter-individual and intra-individual/inter-experiment random variations, which can solve the problem that small animal such as fish cannot be continuously sampled [18]. Within the field of aquaculture medicine, population pharmacokinetics has gained popularity, and extensive research has been conducted on aquaculture animals, including rainbow trout [19], Yellow River carp [18], tilapia [20], giant freshwater prawns [21], and turtles [22]. Unfortunately, there are no similar results for difloxacin in crucian carp.

Therefore, the purpose of this study was to investigate the population pharmacokinetics of difloxacin in crucian carp following a single oral administration at 20 mg/kg body weight (BW). This was achieved using a sparse sampling method and non-linear mixed effect modeling. Additionally, the study aimed to determine the values of PK/PD parameters (free AUC/MIC) using MIC data for susceptible pathogens.

## 2. Materials and Methods

### 2.1. Drugs and Reagents

The analytical standard for difloxacin hydrochloride (C21H19F2N3O3·HCl; Lot No. H0401204) with a purity of 99.6% was obtained from the China Veterinary Drug Inspection Institute (Beijing, China). The commercially available difloxacin hydrochloride powder (100 g: 5 g; calculated as pure difloxacin; Lot No. 22030601) was supplied by Henan Aobang Biotechnology Co., Ltd. (Kaifeng, China). HPLC-grade acetonitrile, methanol, tetrabutylammonium bromide, and dichloromethane were purchased from Tianjin Komiou Chemical Reagent Co., Ltd. (Tianjin, China).

### 2.2. Animals

A total of 30 healthy crucian carp, with an average weight of 0.298 kg (ranging between 0.21–0.40 kg), were bought from Mianchi Qinglian River Aquaculture Co., Ltd. (Sanmenxia, China). These fish were randomly divided into 6 equal groups, with one group as the control to provide blank samples. Every group was housed in a rectangular cage measuring 1.3 m × 0.8 m × 0.65 m with continuous aeration. And the water temperature was kept at 21.3 ± 1.2 °C with heat rods. All fish were acclimatized for at least 10 days and fed daily with drug-free pellet dry feed (pellet size 3 mm) produced upon our request by domestic enterprises. All animal experimentation was performed with full adherence to protocols approved by the Henan University of Science and Technology’s Institutional Animal Care and Use Committee (IACUC). We conducted daily water analyses to ensure optimal conditions, with a pH of around 7.4, total ammonia nitrogen levels ≤0.7 mg/L, nitrite nitrogen levels ˂0.07 mg/L, and dissolved oxygen levels exceeding 7.5 mg/L.

### 2.3. Administration and Sampling

An oral solution containing 10 mg/mL of difloxacin was prepared by dissolving the difloxacin hydrochloride powder in 0.9% physiological saline. To achieve precise dosing in each fish, difloxacin was administered via gavage at a rate of 20 mg/kg body weight. As described previously [18], a 1-mL syringe was used to accurately aspirate the drug solution and slowly push the difloxacin solution through a plastic hose from the mouth of the fish into the belly. To prevent possible reflux, each fish was restrained for about 20 s after administration.

Blood samples (approximately 1 mL) were collected from the fishtail vein at scheduled intervals of 0.25, 0.5, 1, 2, 4, 6, 8, 12, 16, 24, 36, 48, 72, 96, and 120 h after administration. The collected blood was then immediately placed in test tubes containing heparin. Table 1 shows the schedule of blood collection [18]. Each fish was sparsely sampled three times (Table 1). The plasma samples were collected via centrifugation at 3000× *g* for 10 min and were then stored at −20 °C until further analysis.

### 2.4. Determination of Drug Concentrations

To extract difloxacin from plasma, 400 μL of the thawed sample was mixed with 2 mL of dichloromethane and vortexed for one minute, then subjected to centrifugation at 4000× *g* for 10 min. The supernatant was collected in a glass test tube, and the extraction process was repeated. All extracts were collected and evaporated under a nitrogen stream at 60 °C. The residue was reconstituted with 400 μL of a mobile phase, vortexed for 1 min, then centrifuged at 10,000× *g* for 10 min. After this, 20 μL of the supernatant was injected into the HPLC system (Waters Corporation, Shanghai, China), which included an e2695 separations module and a 2487 ultraviolet detector. Chromatographic separation was performed with a reverse-phase C18 analytical column (Hypersil BDS-C18 column 4.6 × 250 mm, 5 μm, Elite analytical instruments Co., Ltd., Dalian, China), which was retained at 40 °C. The mobile phase was 90% tetrabutylammonium bromide buffer (0.03 mol/L; pH = 3.1) and 10% acetonitrile with a flow rate of 1 mL/min. The detection wavelength was 276 nm.

A stock solution of difloxacin with a concentration of 100 μg/mL was prepared in methanol, which was stable at −20 °C for two months. This stock solution was further diluted in the mobile phase to a series of working solutions with concentrations ranging from 1 to 50 μg/mL. These working solutions were spiked into the blank plasma samples to prepare calibration curve samples whose difloxacin concentrations ranged from 0.1 to 5 μg/mL. For precision and accuracy, five replicates at three different concentrations (0.1, 1, and 5 μg/mL) were tested to evaluate the coefficients of variation and recoveries, respectively. The limits of quantification (LOQ) and detection (LOD) were determined based on signal-to-noise ratios of ≥10 and ≥3, respectively.

### 2.5. Population Pharmacokinetic Modeling

This process involved the analysis of plasma concentration data using a non-compartment model analysis (NCA) via Phoenix software (version 8.1; Pharsight, Cary, NC, USA). Naïve average data (NAD) were first used to calculate the values of the apparent volume of distribution per bioavailability (V/F) and clearance rate per bioavailability (CL/F), which served as the initial values for the basic population model [23]. A nonlinear mixed effect model (NLME) was then used to model the plasma concentration data using Phoenix software (version 8.1; Pharsight, Cary, NC, USA). The optimal population model was chosen by comparing comprehensive factors such as the −2LL (−2 × log-likelihood function value), AIC (Akaike information criterion), BIC (Bayesian information criterion), return code, and goodness-of-fit graph [24]. The study utilized the shotgun method to explore the potential correlation between the covariate (body weight) and pharmacokinetic parameters, including CL/F, V/F, and the absorption rate constant (Ka). The changes in ∆−2LL among different scenarios, coupled with the goodness-of-fit graph, were used to evaluate the impact of the covariate on the population model. The result showed that the value of −2LL changed little, and the ∆−2LL value was less than 6.64. Therefore, the covariate was excluded from our model. Furthermore, the relationship between covariances (ηV, ηCL, and ηKa) was also taken into account, and a covariance model was incorporated to establish the final population pharmacokinetic model in which the type of covariance model was a full variance–covariance matrix. The visual prediction check (VPC) method was used to verify the predictive ability of the final model, and the simulation frequency was set to 1000 times. In both the basic and final population models, the log-additive error model was used to obtain precise predictions.

Secondary parameters (K_10_, T_½_, and AUC_0-inf_) for the population were calculated through the final model based on the following formulas:K10 = tvCL/tvV(1)
T_½_ = ln2 × tvV/tvCL(2)
AUC_0-inf_ = Dose/tvCL(3)
where tvCL and tvV were the typical values of the clearance per bioavailability and apparent volume of distribution per bioavailability, respectively; dose was the orally administered difloxacin (20 mg/kg BW); and K_10_, T_1/2_, and AUC_0-inf_ were the elimination rate constant, elimination half-life, and the area under the plasma concentration-time curve from zero to infinity time, respectively. In the population pharmacokinetic modeling process, we utilized Phoenix NLME software (version 8.1; Pharsight, Cary, NC, USA). This powerful tool served multiple purposes, including conducting concentration simulations, calculating PK parameters, comparing different population models, and generating concentration figures.

### 2.6. Statistical Analysis

While constructing the population model, the maximum likelihood ratio test (LRT), LRT = ∆−2LL conforming to Chi-square distribution, was used to differentiate the predictive performances of different models. If the LRT value exceeds the critical value of 6.64, we can conclude that there is a significant difference between the two models at a significance level of 0.01 (*p* < 0.01). Similarly, if the LRT value exceeds the critical value of 10.83, we can conclude that there is a very significant difference between the two models at a significance level of 0.001 (*p* < 0.001) [25].

## 3. Results

### 3.1. Validation of the Analytical Method

The assay method proved to be highly selective for difloxacin and free-from interferences. Additionally, the difloxacin concentrations were linear, in the range of 0.1 to 5 μg/mL. The standard curve (Y = 5 × 10^6^X − 0.0003, R^2^ = 0.9999) was reliable in determining the difloxacin concentration (where Y represents the difloxacin concentration and X represents the peak area). The limit of quantification was determined to be 0.1 μg/mL. The current results show that the average recovery rate was 81.46%, and the intra-day coefficients of variation ranged from 0.74% to 7.72%. Furthermore, the inter-day coefficients of variation varied between 4.81% and 7.66%.

### 3.2. Pharmacokinetic Results after Naïve Averaged Analysis

Difloxacin remained detectable for up to 120 h following a single oral dose of 20 mg/kg BW. The pharmacokinetic parameters obtained via NCA using NAD are shown in Table 2, and a graph demonstrating the individual fits is shown in Figure 1. Only estimates were obtained without any variance information.

### 3.3. Population Pharmacokinetic Comparisons

To develop the basic population model, we used the NAD-calculated V/F and CL/F values (Table 2), analyzed via the non-compartmental method. The resulting model determined the values of tvKa, tvV, and tvCL to be 1.16 1/h, 14.34 L/kg, and 0.18 L/h/kg, respectively (Table 3). Then BW was incorporated into the basic population model as a covariate. However, no significant impacts were found on the pharmacokinetic parameters, and the ∆−2LL value was found to be 3.02. Therefore, the covariate model was not included in the present manuscript. Finally, the covariance model was developed by incorporating the relationship between covariances (ηV, ηCL, and ηKa). Although the RSE% values of tvV, tvKa, and stdev0 were comparable to those in the basic population model, the comprehensive factors such as −2LL and AIC proved that the covariance model was the optimal one. In the final model, the values of tvKa, tvV, and tvCL were 1.18 1/h, 14.18 L/kg, and 0.20 L/h/kg, respectively. By comparing the −2LL, AIC, and BIC values and some goodness plots of basic and final models, it was found that there were very significant differences (*p* < 0.001) between them. Comparisons of the pharmacokinetics variables between basic and final population models are shown in Table 3.

The goodness-of-fit diagram (Figure 1 and Figure 2) demonstrates the accuracy of the final model. Figure 2 shows both observed and predicted concentrations distributed near y = x, indicating a well-fitting model with the predicted value close to the observed value.

Figure 3 displays the absolute values of the CWRES in a symmetric red curve around y = 0. The CWRES data trend fluctuates around y = 0 in a blue curve, and the distribution of the CWRES data between y = ±2 (95% confidence interval) indicates that the CWRES now conform to a normal distribution. However, due to the high plasma concentrations of crucian carp in the last few points, we chose to retain the data and respect the experimental facts that the late time points could not be well predicted.

### 3.4. Final Model Validation

Figure 4 displays the validation results of the final model from the VPC method (1000 simulations). The 5%, 50%, and 95% quantile curve trends of the observed and simulated values demonstrate a common direction. These results suggest that the model error is minimal, and the fitting is excellent. The simulated values were used to determine the 90% confidence interval for the 5%, 50%, and 95% quantile lines. It can be observed that the observed concentrations mostly fall within the quantile range of the simulated values. The shaded portion indicates that the model has good predictive abilities for low and medium concentrations. However, due to the high concentration at the last time point, we chose to respect the experimental data, so the predictive ability is biased at the last time point.

Based on the final population model, the secondary pharmacokinetic parameters for the population were determined (Table 4).

## 4. Discussion

The present report offers the inaugural study on the population pharmacokinetic modeling of difloxacin in crucian carp. We obtained the pharmacokinetic parameters for difloxacin through a non-compartmental analysis using NAD. Nonetheless, the non-compartmental analysis method failed to provide accurate predictions or SD values for the pharmacokinetic parameters. Accordingly, we employed the non-compartmental analysis results (Table 2) solely as initial values to develop a population pharmacokinetic model. In doing so, we were able to refine and enhance our understanding of the pharmacokinetic profile of difloxacin in crucian carp.

We employed the shotgun method in screening potential covariates to develop and optimize population models. Gender identification in non-lethal experiments, like pharmacokinetics, can be challenging for small species like crucian carp, so we only considered BW as a covariate. However, our analysis demonstrated that BW did not significantly impact population pharmacokinetic parameters; therefore, we did not incorporate a covariate model in our study. This contrasted with previous studies [18,19,20] that found weight to have a significant effect on pharmacokinetic parameters.

The final model was obtained by incorporating the relationship between covariances into the basic population model. Compared with the basic population model, similar RSE% values were observed for the parameters of tvKa, tvV, and tvCL. However, a huge variability between fish was observed with the higher ω^2^ values (Table 3). The Δ–2LL value between basic and covariance models was found to be 42.72, indicating a very significant difference between both models (*p <* 0.001) [25]. This significant difference might be caused by random effects [26].

To verify the predictive ability of the final population model, we used the VPC method. For the most part, the difloxacin concentrations were accurately predicted, but the last time point showed some deviation. This could be due to a delay in drug clearance leading to a slower decrease in concentration over time or variability in the measurement techniques used. Future studies should address these factors to improve the accuracy of the concentration measurements. Based on the final population model, the values of tvKa, tvV, and tvCl were determined to be 1.18 1/h, 14.18 L/kg, and 0.20 L/h/kg, respectively. These values were all smaller than the corresponding values for danofloxacin in Yellow River carp, which were 2.48 1/h, 47.8 L/kg, and 0.694 L/h/kg, respectively [18]. These differences indicated that difloxacin had a slower absorption rate, narrower distribution, and slower clearance rate than danofloxacin.

Determining the elimination half-life (T_½_) is crucial for understanding drug elimination rates in animals. Our study found that the T_½_ of difloxacin in crucian carp reared at 21.3 °C was 50.27 ± 4.45 (SD) hours, which was shorter than the elimination half-lives in grass carp (69.47 h; [14]; 15 °C), turbot (94.72 h; [27]; 14–17 °C), and Chinese mitten crab (96.31 h; [28]; 25 °C), indicating a quicker elimination in crucian carp (21.3 °C). Water temperature is a crucial factor in drug elimination rates [29] as lower temperatures led to slower elimination in prior research conducted with crucian carp, and the T_½_ values of difloxacin were found to be 95.36 and 48.93 h at 10 °C and 20 °C, respectively [1].

The AUC is a vital parameter reflecting internal exposure. Our study found that a 20 mg/kg BW oral dose yielded a median AUC of 102.05 h·μg/mL (Table 4). This result was higher than that seen in grass carps at 15 °C (81.55 h·μg/mL) [14] but lower than in turbot (246.66 h·μg/mL at 14–17 °C) [27] and Chinese mitten crab (500.907 h·μg/mL at 25 °C) [28]. Although the same dose was administered in these studies, differences in rearing temperatures or animal species may explain these variations. In another crucian carp breed (*Carassais auratus gibebio*) [30], AUC values were higher at lower temperatures. At 16 °C, for example, the AUC was 763.761 h·μg/mL—much greater than the 243.244 h·μg/mL at 25 °C. The slower blood flow and lower metabolic rate induced by cold temperatures likely increased the internal exposure to difloxacin. Only one temperature (21.3 °C) was applied in the current study. Therefore, population pharmacokinetics at other water temperatures should be carried out in the future.

As concentration-dependent antibacterials, fluoroquinolones have been shown to have a PK/PD index with a free AUC/MIC ratio ≥125, which has been effective in achieving clinical and bacteriological success, as well as in preventing the emergence of resistance [31]. The plasma protein-binding levels of difloxacin have been reported in the range of 52.66% to 80.56% in healthy gibel carp [9]. If we assumed an average value (66.7%) of the protein-binding rate in crucian carp, the current median free AUC might be 33.98 h·μg/mL. The MIC data of difloxacin against *Aeromonas hydrophila* isolated from carps were reported to be 0.83 to 4 μg/mL [9]. Assuming that the current MIC ranges remain the same, the free AUC/MIC ratios were calculated to be above 8.495 and 40.93 against *Aeromonas hydrophila*. This means that the current 20 mg/kg oral dose of difloxacin might not be enough to treat infections in crucian carp. However, further study should be carried out to collect the pathogens from crucian carp and determine the MIC data of difloxacin against them. Pharmacodynamic experiments must also be further carried out to determine the optimal therapeutic dose for the treatment of *Aeromonas hydrophila* infection.

## 5. Conclusions

In this study, we found that difloxacin has favorable pharmacokinetic properties in crucian carp, with quick absorption and slow elimination rates. We utilized sparse sampling and non-linear mixed effect modeling methods to accurately predict its pharmacokinetics. By adding the relationship between covariances to the final population model, similar RSE% values were observed for the tvKa, tvV, and tvCL. However, our results show that the current oral dosing of difloxacin (20 mg/kg BW) might not be enough to inhibit pathogens with MIC values above 0.83 μg/mL, as calculated by the AUC/MIC ratio. More research is necessary to establish the effectiveness of difloxacin in combatting the latest pathogens isolated from crucian carp. Moreover, it is essential to evaluate the impact of temperature fluctuations on the pharmacokinetics of difloxacin in these fish.

## Figures and Tables

**Figure 1 vetsci-10-00416-f001:**
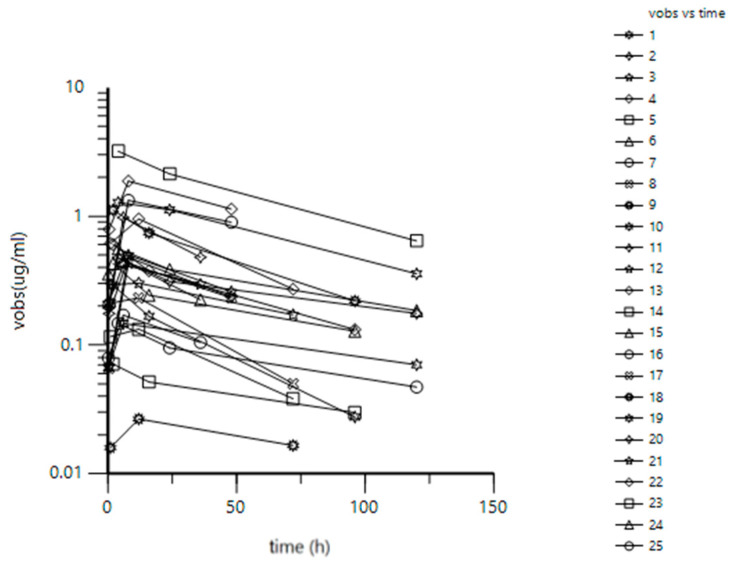
Individual concentration data presented as a spaghetti plot after a single oral dose of difloxacin at 20 mg/kg BW.

**Figure 2 vetsci-10-00416-f002:**
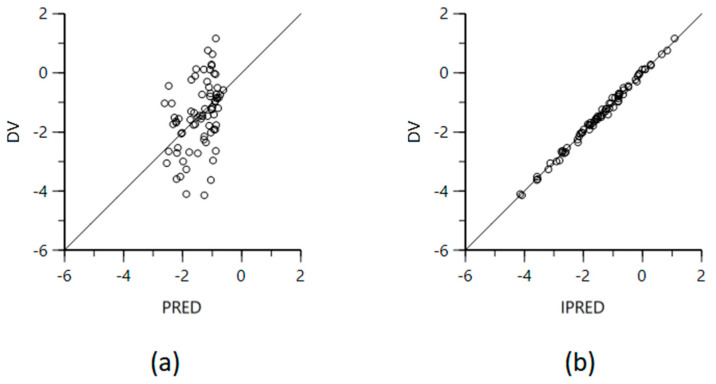
(**a**) Logarithmic plot of the dependent variable (DV, difloxacin concentrations) versus population predicted difloxacin concentrations (PRED) and (**b**) logarithmic plot of the dependent variable (DV, difloxacin concentrations) versus individual predicted difloxacin concentrations (IPRED). Open circles indicate difloxacin concentrations and solid line repesents unity.

**Figure 3 vetsci-10-00416-f003:**
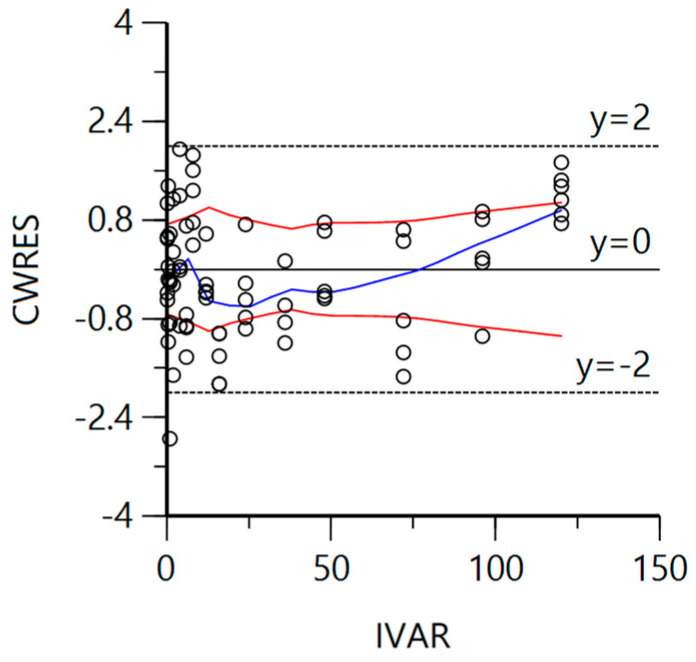
The diagnostic plot of the goodness-of-fit of the final model in which CWRES represents the conditional weighted residuals, and IVAR (independent variable) represents time (hours).

**Figure 4 vetsci-10-00416-f004:**
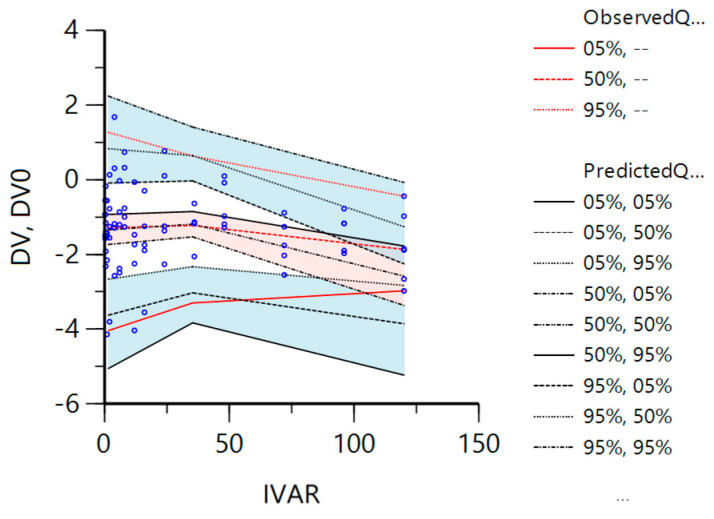
The validation results of the final model from the VPC method. The blue dots are observed concentration data, the *x*-axis (IVAR) is the time (h), the *y*-axis represents the dependent variable (difloxacin concentration; μg/mL), and DV is predicted concentrations, while DV0 is the observed ones. The three red lines of Observed Q represent the 5%, 50%, and 95% quantiles of observed concentrations, and the three black lines of Predicted Q represent the 5%, 50%, and 95% quantiles of predicted ones, the blue and red areas represent the confidence intervals (with a level of 90%) around the 5%, 95%, and 50% quantiles of predicted concentrations, respectively.

**Table 1 vetsci-10-00416-t001:** Blood collection time (hour) points from each crucian carp after a single oral administration of difloxacin.

ID	BW (kg)	Blood Collection Time (Hour) after the Administration														
		0.25	0.5	1	2	4	6	8	12	16	24	36	48	72	96	120
1	0.21						*					*				
2	0.33		*					*					*			
3	0.37			*					*					*		
4	0.35				*					*					*	
5	0.31					*					*					*
6	0.26	*					*					*				
7	0.28		*					*					*			
8	0.26			*					*					*		
9	0.27				*					*					*	
10	0.27					*					*					*
11	0.30	*					*					*				
12	0.27		*					*					*			
13	0.34			*					*					*		
14	0.23				*					*					*	
15	0.40					*					*					*
16	0.25	*					*					*				
17	0.28		*					*					*			
18	0.27			*					*					*		
19	0.29				*					*					*	
20	0.36					*					*					*
21	0.33	*					*					*				
22	0.29		*					*					*			
23	0.29			*					*					*		
24	0.34				*					*					*	
25	0.31					*					*					*

The * symbol indicates that this individual was sampled at this time point.

**Table 2 vetsci-10-00416-t002:** Pharmacokinetic parameters of difloxacin obtained via non-compartmental analysis using NAD.

Parameters	Units	Estimate
AUC_0-t_	h·μg/mL	39.67
AUC_0-inf_	h·μg/mL	60.14
V	L/kg	27.63
CL	L/h/kg	0.03
T_max_	h	4.00
C_max_	μg/mL	1.13
T_½_	h	57.59
%AUC	%	34.03

AUC_0-t_—the area under the plasma concentration–time curve from time 0 to the last time point; AUC_0-inf_—the area under the plasma concentration–time curve from 0 to infinity; V—apparent volume of distribution per bioavailability; CL—total clearance per bioavailability; T_max_—time to reach peak concentration; C_max_—peak concentration; T_½_—elimination half-life; %AUC—percentage of the area under the concentration–time curve that has been derived after extrapolation.

**Table 3 vetsci-10-00416-t003:** Parameter estimations based on basic and final population pharmacokinetic models.

Model	Parameter	Units	Estimates	RSE%	2.5%CI–97.5%CI	Inter-Individual Variation ω^2^ (RSE%)
Basic model (No covariance model)	tvKa	1/h	1.16	14.22	0.83 to 1.50	0.60 (39.14)
	tvV	L/kg	14.34	20.85	8.37 to 20.30	1.16 (31.52)
	tvCL	L/h/kg	0.18	18.02	0.12 to 0.25	0.81 (27.52)
	stdev0	μg/mL	0.19	11.88	0.15 to 0.24	/
Final model (Covariance model)	tvKa	1/h	1.18	12.55	0.88 to 1.47	0.69 (35.14)
	tvV	L/kg	14.18	18.88	8.83 to 19.53	1.12 (28.54)
	tvCL	L/h/kg	0.20	17.96	0.13 to 0.27	0.98 (27.08)
	stdev0	μg/mL	0.16	11.06	0.12 to 0.19	/

The symbol tv stands for “typical value”, Ka represents the absorption rate constant, V denotes the apparent volume of distribution per bioavailability, and CL signifies the total body clearance per bioavailability. The stdev0 parameter is the random-error term describing variations in difloxacin concentration in crucian carp plasma, as modeled in this study. RSE% is the coefficient of variation, and / indicates not applicable.

**Table 4 vetsci-10-00416-t004:** Secondary pharmacokinetic parameter values derived from the final model.

Secondary Parameters	Units	Estimates	RSE%	2.5%CI	97.5%CI
K_10_	1/h	0.01	8.71	0.01	0.02
T_½_	h	50.14	8.71	41.42	58.86
AUC_0-inf_	h·μg/mL	102.05	17.96	65.44	138.67

K_10_—elimination rate constant; T_½_—elimination half-life; AUC_0-inf_—the area under the plasma concentration–time curve from zero to infinity time.

## Data Availability

The data that support the study findings are available upon request and after authorization by the authors.

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
