# Peer review of "Population Pharmacokinetics of Difloxacin in Crucian Carp (*Carassius auratus*) after a Single Oral Administration"

_vetsci, 2023, doi:10.3390/vetsci10070416_

Round 1

Reviewer 1 Report

Comments to the authors

The population pharmacokinetics models have been developed for difloxacin in crucian carp (Carassius carassius), and the covariate of body weight and random effects were sequentially incorporated into the population model. And the final population model was determined by comparing comprehensive factors and goodness of fit graph. Based on the final model, the population pharmacokinetic parameters were determined and the corresponding PK/PD index was calculated.

The manuscript contains new data and its contribution to the field is significant. English language is at an acceptable level. The manuscript can be accepted for publication.

My main concerns are:

1.     Is 30 animals enough for a population pharmacokinetic study?

2.     Since population model was developed based on the nonlinear mixed effects approach, are the authors willing to explain how tvKa is calculated in the paper?

Other suggestions:

1. It is recommended to italicize all scientific names of organisms, including the crucian carp (Carassius carassius).

2. Line 10: Is there Affiliation 2? If no, please removed this line.

3. Line 37: Based on the calculated PK/PD index of AUC/MIC.

4. Lines 120-125: At each time point, approximately how many blood samples were collected?

5. Line 170: The suffixes of the three secondary parameters should be indicated by subscripts.

6. Line 184: R2, the 2 should be a subscript.

7. Table 2: Please use subscripts correctly in the notes to Table 2.

8. Table 3 can be optimized. Look at line spacing.

9. The resolution of Figure 2 is too low, please replace the figure with a higher resolution.

10. Lines 264-265: Is there any precedent for using sex as a covariate in previous fish population pharmacokinetic models? Or any other factors were identified as one covariate in fish population pharmacokinetic models?

11. Line 304: The two pathogens should be in italics.

12. Line 320: 6. Patents?

good.

Author Response

Dear Reviewer 1,

We are truly grateful for your positive and constructive comments and suggestions on our manuscript entitled “Population pharmacokinetics of difloxacin in crucian carp (Carassius auratus) after one single oral administration”. Based on these comments and suggestions, we have carefully modified the original manuscript. All changes made to the manuscript are marked up using the Track Changes function.

Our point-to-point responses to the comments are attached to this mail. We hope this revision meets Veterinary Sciences’ standards and receives some positive comments.

Yours sincerely,

Fan

Reviewer 2 Report

The manuscript describes the application of population pharmacokinetics (PK) to determine the PK parameters of difloxacin in crucian carp along with the variations of the estimated parameters. Because research on population PK is relatively rare in the field of fish pharmacology, this manuscript is helpful to demonstrate the benefits and applicability of population PK for antimicrobial therapy in fish. In general, the manuscript was concise and well-organized. The objective and research gap were clearly written. The manuscript deems acceptable for publication after minor revision. Here, I have a few comments for the authors.

1.    Line 10: Please remove “Affiliation 2; e-mail@e-mail.com”.
2.    Line 58-59: It is unclear what is the uniqueness of difloxacin compared to other fluoroquinolones such as enrofloxacin. Please provide more description with references.
3.    Line 62-63: According to the reference No. 7 (Water animal cultivation medication instruction booklet No. 1 and No. 2.), it appears that difloxacin is not on the approved drug list in China. Please recheck this information.
4.    Line 193: Please remove “3.1. Subsection”.
5.    Line 279: The sentence “difloxacin had faster absorption, narrower distribution, and slower clearance than danofloxacin” is not true since the absorption rate constant (Ka) of difloxacin (1.19 1/h) was lower than that of danofloxacin (2.48 1/h).
6.    Line 289: The reference No. 22 seems inappropriate in this context because this paper did not discuss the meaning of AUC.
7.    Line 304-306: The reference No. 24 (Wang and Zhu, 1992) for the MIC values of 0.05-0.2 for A. hydrophila and 0.025-0.05 for A. sobria was too old. The current MICs may be much different from the previous data. So, it is preferable to cite more recent publications. In addition to the target MIC of 0.2 ppm (Line 309), it could be very helpful if the authors also use theoretical MIC that may be found in a real situation such as MICs of 1 and 2 ppm. It is interesting to know what would be the optimal dosages for the target MICs of 1 and 2 ppm when determined from the population PK approach. Finally, the method of optimal dose determination by population PK approach should also be given in the Materials and Methods section, and present the determined optimal dosages in the Results section.
8.    Line 320: Please remove “6. Patents”.

Author Response

Dear Reviewer 2,

We are truly grateful for your positive and constructive comments and suggestions on our manuscript entitled “Population pharmacokinetics of difloxacin in crucian carp (Carassius auratus) after one single oral administration”. Based on these comments and suggestions, we have carefully modified the original manuscript. All changes made to the manuscript are marked up using the Track Changes function.

Our point-to-point responses to the comments are attached to this mail. We hope this revision meets Veterinary Sciences’ standards and receives some positive comments.

Yours sincerely,

Fan

Reviewer 3 Report

This study examined the pharmacokinetics of difloxacin in crucian carp (Carassius auratus) and established a population pharmacokinetic model using nonlinear mixed effects method. Findings suggest that body weight is not a significant factor in population PK parameters, and the final model provides accurate predictions. The study also determines an adequate oral dose of difloxacin (20 mg/kg body weight) that effectively inhibits pathogens with MIC values below 0.02 μg/mL. With some minor issues to be addressed, this paper holds great potential for publication.

Specific comments:

1. Lines 3, 12, 26, 43: The Latin scientific name should be italicized.

2. Line 67-77: Traditional pharmacokinetic studies in aquatic animals often involve a high volume of subjects but yield limited profiles. This widely accepted consensus should be supplemented in this paragraph.

3. Line 116: Why did the authors choose the administration route by gavage rather than the mediated feed?

4. Lines 193: “3.1. Subsection”Remove

5. Adjust the appropriate column width in Table 3, with 'Parameter' as much as possible in one row. Centered alignment in Figure 1?

6. Line 221: This paragraph is not indented on the first line and is aligned at both ends.

7. Line 252: Missing a period.

8. Line 295: “Carassais auratus gibebio” use italics.

9. Lines 304, 307, and 308: “Aeromonas hydrophila” and ”Aeromonas sobria” use italics.

Please check spelling, grammar, and punctuation.

Author Response

Dear Reviewer 3,

We are truly grateful for your positive and constructive comments and suggestions on our manuscript entitled “Population pharmacokinetics of difloxacin in crucian carp (Carassius auratus) after one single oral administration”. Based on these comments and suggestions, we have carefully modified the original manuscript. All changes made to the manuscript are marked up using the Track Changes function.

Our point-to-point responses to the comments are attached to this mail. We hope this revision meets Veterinary Sciences’ standards and receives some positive comments.

Yours sincerely,

Fan

Reviewer 4 Report

The manuscript is well written, and a wide range of factors have been considered to study the population PK of difloxacin in crucian carp.

To further improve the quality of the work, I suggest the following:

The authors in the discussion (lines: 266-267) write that “However, our analysis demonstrated that BW did not significantly impact population pharmacokinetic parameters, and therefore, we did not incorporate a covariate model in our study.”  The authors should explain why the BW did not impact the PPK parameters, because as they known the dose is express as mg/kg and for many dugs the PK depend from BW and gender of the animal species.

The authors should explain why the experiment take place at 21.30C water temperature and why they use carp with an average weight of 0.298 kg and if this condition of the experiment has any impact on the PK behavior of the drug.  

The authors must be rewriting the line 270-280 of the discussion using more formal English.

Author Response

The manuscript is well written, and a wide range of factors have been considered to study the population PK of difloxacin in crucian carp.

Response: Thanks very much for your affirmation.

To further improve the quality of the work, I suggest the following:

The authors in the discussion (lines: 266-267) write that “However, our analysis demonstrated that BW did not significantly impact population pharmacokinetic parameters, and therefore, we did not incorporate a covariate model in our study.”  The authors should explain why the BW did not impact the PPK parameters, because as they known the dose is express as mg/kg and for many dugs the PK depend from BW and gender of the animal species.

Response: Thanks for your question. Based on the valuable comments from the other reviewers, we think the current experimental design may be the main reason for this result. Based on the consideration of animal welfare, we adopted a sparse sampling method, and only 3 plasma samples were collected from each fish.

The authors should explain why the experiment take place at 21.30C water temperature and why they use carp with an average weight of 0.298 kg and if this condition of the experiment has any impact on the PK behavior of the drug.

Response: Thanks for your questions. Regarding the water temperature, we had a plan to maintain the water temperature at 20 degrees Celsius using a heating rod, but the actual measurement results showed that the average water temperature was 21.3 ± 1.2°C, which was slightly higher than the planned temperature. However, this temperature is also the survival temperature for crucian carp. In terms of weight, we chose medium-sized crucian carps with an average weight of 0.298 kg, which are convenient for feeding and plasma collections. Thanks for your question. We will conduct further studies in carps with different weights and / or reared at different temperatures. We believe that the temperature effect must exist, but whether the body weight effect exists still needs to be confirmed by the results of further experiments.

Reviewer 5 Report

·         General comments

This article presents a PK study of danofloxacin in crucian carp after a single oral administration, by using a population PK modeling approach. Although the use of popPK model should be encouraged, this article has major flaws and confusions regarding the methodology that need to be carefully revised.

Moreover, there are many references that are not relevant (self-citation) and/or that do not support/justify the ideas.

  • Specific comments 

L22 : it is not recent MIC data. Please correct

L23 and L35-36 : Please be more prudent about your statement (not using can, but rather could/may/might).

Be careful about the MIC value that you provide as a threshold: is it 0.2 or 0.02 µg/mL ? Review carefully all the manuscript for that point.

L43 and 50 : references 1 and 3 are not relevant to support your statement

L57 : what does “similar effect” means. In this section, please give more detail about the advantage of using difloxacin compared to the other fluoroquinolones that are already used in fish production.

L120 : what was the volume of each blood sample ?

L155 : use CL/F and V/F (instead of CL and V) to highlight that it is apparent parameter, and also in Table 2

L157 : use abbreviation NLME instead of NONMEM which is the name of a famous software

L160 : what does “return code” mean ? Also reference 17 is not relevant, too specific for covariate analysis

L160-169 : Please carefully revised your methodology which is confusing.

What do you mean by random effect ? Is it only the IIV or something else ? Because it seems that your basic PK model already includes IIV (therefore random effect).

Explain the type of relationship (with equation/formula) you explored between BW and PK parameters. And explain clearly how this relationship was assessed, because “significant impacts” is unclear.

Explain the type of covariance model that you have explored ? Is it a full variance-covariance matrix ?

Have you got data < LOQ ? If yes, how did you handle it in the pop PK model ?

Section 3.2

You must provide a graph presenting the raw PK data (in a log scale), so that the reader can observe the PK profile.

L193 : Remove the end of your sentence (3.1 subsection)

In table 2 : add the % AUC extrapolation between AUC0-t and AUC 0-inf

Section 3.3 : all this section must be revised in accordance with the correction of methodology of section 2.5.

Moreover you must add a graph with the individual fits

L201 : use CL/F and V/F

L205 : what does “significant CV% values “mean ?

L209-210 : this seems wrong. When looking at Table 3, the CV% were not all decreased in the final model. Or be more precise about which CV% you referred to

Overall Table 3 is confusing :

- column 5 (CV%) referred to the precision of estimation. It should be named RSE % (relative standard error). And this is redundant with the value of SE in the Column estimates. Keep only either SE, either RSE.

- carefully revise the values of the CI%, for instance there is an error for tvV of basic model (not possible 2.01).

- for the last Column, again the CV% must be named as RSE% to avoid confusion with the value of omega.

- give the values of the correlation between random effect (covariate model). Is the covariate model really relevant ? because the omega values increased for Ka and CL between basic and final model

- Precise if the error model is additive, proportional or else

- Figure 1 : Add the graph DV vs PRED (not only IPRED). The graph on the right show misspecification for the late time points. This should be discussed

Section 3.4

- Fig 2 : no need of 2 panels, the right one is sufficient. Modify the binning strategy used for plotting as the early time points are not included in the prediction.

Your VPC show wide PIs which means that the variability is overestimated. With VPCs 90%, you should observe 10% of data outside the prediction interval which is not the case  This is obvious when looking at the value of omega in Table 3. Moreover, again the last time points is not well predicted, you have a major bias in your VPC.

Some IIV may be excluded from your model, because due to the sparse sampling method, there may be not enough information to support relevant estimation of IIV for all parameters.

Overall, please check your modelling process and adjust your final model.

- Table 4 : again the CV% reflects the precision of estimation, should be tagged as RSE and this is redundant with the SE column.

Section Discussion

- L259-260 : explain why the NCA did not give accurate prediction ? Your sampling design is likely to be an explanation. You have not sampled over a long enough period of time.

- L266 : again, the BW relationship with PK parameters was not found but it may be due to your experimental design.

- L270-271  and L274: Still confusing the way you use “random effect”. Please clarify

- L283 : Is it mean +/- SD or +/- SE ? this SD should be used

- L289 : reference 22 is not relevant with your statement

- L290-297 : when you compare AUC values, is it base don the same oral dose (20 mg/kg)?. State it or correct the AUC by the doses if you want to compare them.

- L303 : reference 13 is not relevant with your statement

Please discuss the value of AUC/MIC > 125 as it was historically performed with bacteria from mammals grown at 37°C. Please check if it is the AUC/MIC of the freeAUC/MIC that needs to be over 125.

-L305 : State clearly that the MIC distribution are old data and be more moderate about your affirmation that danofloxacin is effective against these bacteria.

Author Response

Dear Reviewer 5,

We are truly grateful for your positive and constructive comments and suggestions on our manuscript entitled “Population pharmacokinetics of difloxacin in crucian carp (Carassius auratus) after one single oral administration”. Based on these comments and suggestions, we have carefully modified the original manuscript. All changes made to the manuscript are marked up using the Track Changes function.

Our point-to-point responses to the comments are attached to this mail. We hope this revision meets Veterinary Sciences’ standards and receives some positive comments.

Yours sincerely,

Fan

Round 2

Reviewer 5 Report

Thanks for having taken into account most of my previous comments but I was unpleased to see that you still have a lot of unrelevant references, even new ones. Please be very attentive of the choice of the references.

I still have some comments

Ref 1-4 are articles about PK in fish. They do not support at all your general statements from line 46 to line 50.

Line 64 : Please give reference to support the statement of extra-label use

Line 67 : reference 11 not relevant

Line 83 : ref 17 not relevant as it concerns humans!

Fig1 : these are not the individual fits but the individual data presented as spaghetti plot

You have a huge variabilty between fish which is confirmed with the high omega values. Please discuss it in the discussion section

Fig2 (a) : DV vs PREd must be on the same scale (as you did for IPRED)

L251 : the fitting is not excellent because the late time points are not well fitted.

You have also to discuss it more closely. Because the lite tame points are not well fitted, the popPK model is not optimal therefore the secondary parameters are likely biased (as the T1/2 and The AUC). This may also mean that a 2 compartmental model would be better to describe the PK profils if you would have more late sampling times

Line 320 : ref 19 not relevant

L307 : precise that it it the median AUC

L323: precise that it it the median free AUC for this dosing regimen

L337 : wrong statement, your covariance model has increased the omega values for Ka and CL !

And also you kept the old wrong version of your conclusion with MIC values below 0.02 µg/ml (L338-340). Correct it in accordance with the updtaed calculations made (Line 326)

Overall, you have to discuss more closely the high variability oberved in the individual PK profiles, the consequence of bad fitting of late time points (value of AUC not well estimated as already shown in the NAD analysis), and the need of a longer sampling time to be able to characterize properly the PK of difloxacin in this species

Author Response

Dear Reviewer 5,

We are truly grateful for your positive and constructive comments and suggestions on our manuscript entitled “Population pharmacokinetics of difloxacin in crucian carp (Carassius auratus) after one single oral administration”. Based on these comments and suggestions, we have carefully modified the original manuscript. All changes made to the manuscript are marked up using the Track Changes function.

Our point-to-point responses to the comments are attached to this mail. Because there are few literatures on the application of difloxacin in aquaculture, we are confused by the contradictory comments from you and the editor. Based on your comments, we removed some irrelevant references here. However, the editor asked us to provide a minimum of 30 references.

We hope this revision meets the standards of Veterinary Science and receives some positive comments.

Yours sincerely,

Fan Yang
